# In–silico simultaneous respiratory and circulatory measurement during voluntary breathing, exercise, and mental stress: A computational approach

**Masami Iwamoto** ®*, **Satoko Hirabayashi** ®, **Noritoshi Atsumi** ®

Human Science Research-Domain, Toyota Central R&D Labs., Inc., Nagakute, Japan

* iwamoto@mosk.tytlabs.co.jp

## Abstract

Voluntary breathing (VB), short–term exercise (STE), and mental stress (MS) can modulate breathing rate (BR), heart rate (HR), and blood pressure (BP), thereby affecting human physical and mental state. While existing experimental studies have explored the relationship between VB, STE, or MS and BR, HR, and BP changes, their findings remain fragmented due to individual differences and challenges in simultaneous, BR, HR, and BP measurements. We propose a computational approach for in–silico simultaneous measurements of the physiological values by comprehensive prediction of the respiratory and circulatory system responses to VB, STE, or MS. Our integrated model combines a respiratory system with a circulatory model, leveraging actor–critic reinforcement learning to control respiratory muscles. We introduce specific parameters to account for involuntary or VB and hyperventilation induced by STE. We modeled mental stress as an electrical input to the amygdala based on prior studies indicating that stress leads to amygdala hyperactivity. Our predictions for breathing rate (BR), tidal volume, minute ventilation, HR, and BP are validated against literature data obtained during various conditions, including different VB patterns (ranging from 6 to 14 bpm), active or passive knee flexion STE, and MS load. The model demonstrates good agreement with experimental results and highlights its ability to explore the mechanism of individual differences. Our model predicts heart rate variability (HRV) indices of total power spectral density and the ellipse area of Poincaré plot. Notably, slow deep breathing at a BR of 6 bpm increases HRV indices, promoting relaxation and cognitive performance. Conversely, MS elevates BP but reduces HRV indices, indicating an unstable and risky state for mental and physical health. Overall, our proposed computational approach provides simultaneous and reasonable predictions of various physiological values, accounting for individual variations through specific parameters.

## Author summary

The impact of changes in physical or mental state on physiological parameters, such as heart rate and blood pressure, remains uncertain due to inter–individual variability and

**Data Availability Statement:** The source code and data used to produce the results and analyses presented in this manuscript are available for at

https://osf.io/k46r8/?view_only=
d2fa6f3f571348a1b5c08ce515df594f.

**Funding:** The author(s) received no specific
funding for this work.

**Competing interests:** The authors have declared
that no competing interests exist.

challenges in simultaneous measurements. We propose a novel computational model for
predicting respiratory and circulatory responses to voluntary breathing, short–term exer-
cise, or mental stress. We introduce specific parameters to account for involuntary or vol-
untary breathing and hyperventilation induced by short–term exercise. We modeled
mental stress as an electrical input to the amygdala based on prior studies indicating that
stress leads to amygdala hyperactivity. The model reproduced the respiratory and circula-
tory states during active or passive short–term exercises, voluntary breathing control, and
mental stress loads. The proposed computational approach enables simultaneous and rea-
sonable prediction of diverse physiological values, accounting for individual differences
through specific parameters, during voluntary breathing, short–term exercise, or mental
stress. This approach also enables substantial prediction of cognitive performance and
states for mental and physical health through the physiological values and heart rate vari-
ability indices of total power spectral density and the ellipse area of Poincaré plot.

## Introduction

While humans can voluntarily regulate their respiratory state, heart rate (HR) and blood pres-
sure (BP) remain beyond their conscious control. Voluntary breathing (VB), short-term exer-
cise (STE), and mental stress (MS) have been shown to influence breathing rate (BR), HR, and
BP dynamics [1–6]. Understanding the mechanism underlying these changes is crucial for
promoting overall physical and mental well–being [7–10]. Particularly, emotions interact with
autonomic nervous control [11] and induce cardiovascular changes [12]. Therefore, a better
understanding of the mechanisms by which VB, STE, or MS alter the HR and BP is critical for
enhancing physical and mental health.

VB impacts HR, BP, and heart rate variability (HRV) [2, 3, 13, 14], while MS also influences
the BR, HR, and BP [5, 13, 15, 16], and vice versa [9, 10]. STE induces hyperventilation and
affects HR/BP [1, 4]. Especially, the relationship between minute ventilation (MV) and carbon
dioxide partial pressures ($PaCO_2$) in the blood is critical to understand the respiratory and car-
diovascular responses during STE [17] and increase in MV with dead space, that is, with larger
airway resistance, generates short–term and long–term potentiation of exercise ventilatory
response [18]. However, due to inter–individual variability and challenges in simultaneous
measurements (particularly during physical activity), these studies offer only partial insight.
Despite the ability to measure the physiological parameters, such as BR, HR, BP, and HRV, in
the respiratory–circulatory system, establishing a precise relationship among these variables
remains challenging due to their strong correlations.

Considering physiological values, a computational approach can offer a comprehensive
understanding. Previous studies have developed computational models of the respiratory con-
trol system or cardiovascular model to simulate the effect of dynamic exercise [19, 20]. The
model reported by Waldrop et al. (2011) [19] is a computer-based mathematical model of the
respiratory control system including rapidly active neural mechanisms, short–term potentia-
tion, and serum [$K^+$] in addition to the feedback mechanisms of $PaCO_2$ and oxygen partial
pressures in the blood ($PaO_2$), and generates ventilatory and $PaCO_2$ responses to dynamic
exercise for 6 min. However, it does not include the cardiovascular system and thereby cannot
simulate the responses of HR and BP during exercise. The model reported by Roy et al. (2023)
[20] is also a computer simulation-based mathematical model of the cardiovascular hemody-
namic system including its parameters such as HR, cardiac output, and mean arterial pressure,
and simulates the exercise effect on the cardiac parameters relevant to cardiac rehabilitation.

However, it does not include the respiratory control system and thereby cannot simulate respiratory responses, such as the BR and MV, during exercise.

Given the established connection between emotions or affective states and parameters such as HR, BP, and HRV [11], recent studies have applied deep learning techniques to associate these parameters [21–23]. Deep learning methods leverage substantial experimental data from physiological measurements, including HR and facial expression, often captured in static or passive states, and supplemented by subjective ratings. However, the accuracy of emotion or affective state estimation remains at approximately 70%, which is insufficient [21–23]. This limitation arises due to inter–individual variability and complex interactions among physiological values. Moreover, deep learning algorithm cannot explain the influence of external stimuli on physiological values and emotions.

Recent advancements propose alternative computational approaches to explore the interplay between external stimulation, physiological parameters, and brain activity [24–26]. For example, a dynamic system model utilizing coupled autonomous oscillators that emulate brain and cardiorespiratory dynamics has been employed for emotion recognition. This model demonstrated high recognition accuracy using the circumplex model of affect for arousal and valence [24], suggesting the strong correlations between the cardiorespiratory system and emotion. Similarly, investigations into attention mechanisms revealed a positive relationship between attention and tonic locus coeruleus (LC) frequency entrained by respiration using a similar model [25]. Spiking neural network models have explored the LC–amygdala interaction and cardiorespiratory centers, elucidating the effects of external body vibrations on BR, HR, and arousal levels [26]. However, we lack models capable of predicting BR, HR, BP, and HRV simultaneously during VB, STE, or MS, while also examining the impact of these stimulations on physiological responses.

This study presents a computational approach for in–silico simultaneous measurements of physiological values, such as the BR, tidal volume (TV), MV, $PaO_2$, HR, BP, and HRV during VB, STE, or MS by predicting the responses of the respiratory–circulatory system to the external stimulations, aiming to elucidate the mechanisms altering the physiological values. The validity of this approach is assessed by comparing predictions with experimental data from the literature. Additionally, we explore the influence of VB, STE, or MS on physiological values impacting human physical or mental state. The proposed method reveals the interactions among these physiological values during respiratory–circulatory responses, which are not fully captured by in–vivo physiological measurements or artificial intelligence (AI)–based data analysis requiring extensive datasets. This approach also enhances understanding of the ways in which human activities such as, slow, deep breathing, improve HRV and performance.

## Results

### Physiological changes during active or passive short-term exercises

The first condition models the respiratory and circulatory changes during active and passive exercises, based on a prior study by Ishida et al. (2000) [1]. This study involved 13 healthy young adult males (mean age: 22.9 years) and 13 elderly males (mean age: 66.8 years), who either voluntarily flexed their knees or had their knees passively flexed at a constant velocity by an experimenter. Measurements included the BR, TV, MV, HR, systolic blood pressure (SBP), and diastolic blood pressure (DBP). We adjusted the PONS, RAMPI, UMAXE, TML, EXGAIN, and PCO2 parameters iteratively to replicate the respiratory and circulatory states observed in the young adult subjects [1] (Materials and methods and Table 1). Fig 1 compares the simulation results (red solid lines) with experimental data (blue solid lines) [1] for physiological values over time during active STE involving knee flexions. Yellow rectangles indicate

**Table 1. Parameters to control respiratory states.** A parameter of APSR is set to 1.0 in all conditions.

| Sim. | Conditions | BR [bpm] | Involuntary/voluntary breathing | | | Exercise | | | |
|---|---|---|---|---|---|---|---|---|---|
| | | | PONS | RAMPI | UMAX | UMAXE | TML | EXGAIN | PCO2 |
| 1 | Active exercise | 14.5 | 0.67 | 1.0 | 0.0 | 0.07 | 2.0 | 1.0 | 0.02 |
| | Passive exercise | 14.5 | 0.67 | 2.3 | 0.0 | 0.01 | 2.0 | 0.001 | 0.02 |
| 2 | Respiration A | 14 | 0.69 | 2.1 | 0.05 | 0.0 | 0.0 | 0.0 | 0.0 |
| | Respiration B | 12 | 0.63 | 2.3 | 0.05 | 0.0 | 0.0 | 0.0 | 0.0 |
| | Respiration C | 10 | 0.55 | 2.0 | 0.05 | 0.0 | 0.0 | 0.0 | 0.0 |
| | Respiration D | 8 | 0.40 | 2.5 | 0.05 | 0.0 | 0.0 | 0.0 | 0.0 |
| | Respiration E | 6 | 0.33 | 2.5 | 0.05 | 0.0 | 0.0 | 0.0 | 0.0 |
| 3 | Stress loading | 16 | 0.72 | 1.3 | 0.0 | 0.0 | 0.0 | 0.0 | 0.0 |

the responses during the exercise. The physiological parameters include the BR, TV, MV, HR, and BP. Additionally, Fig 1 also presents the simulation results (red solid line) of the relationship between the heartbeat frequency and power spectral density (PSD) of HRV, noting that experimental HRV data are not reported in the referenced study [1]. The exercise commenced

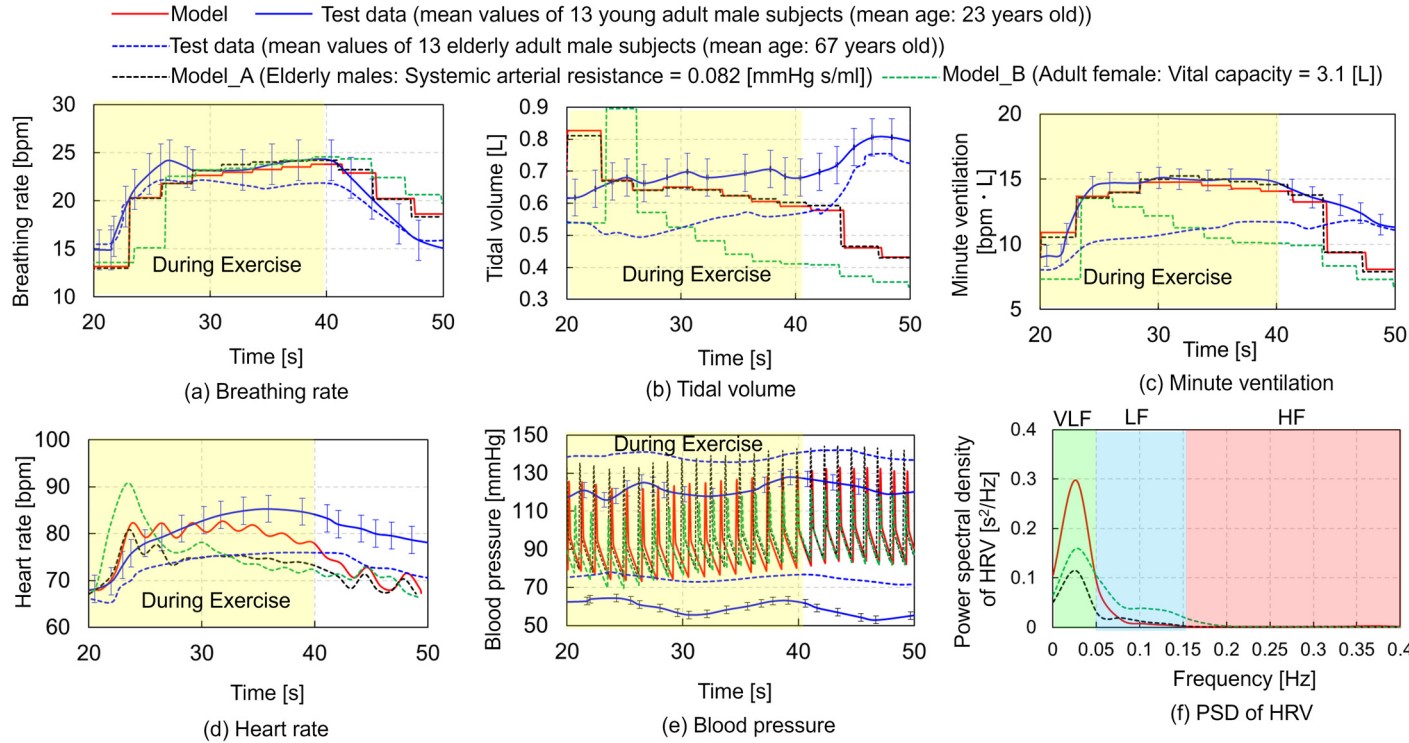

**Fig 1. Comparisons between simulation results (red solid lines) and experimental data of young and elderly adult males (blue solid and dotted lines, respectively) [1] with mean values (solid lines) and standard error (error bars) exclusively for young adult males of physiological values versus time during active short–term exercise involving knee flexions.** The yellow rectangles correspond the responses observed during the exercise. (a) Breathing rate (BR), (b) Tidal volume (TV), (c) Minute ventilation (MV), (d) Heart rate (HR), (e) Blood pressure (BP) vs. time, and (f) Power spectral density (PSD) of heart rate variability (HRV) vs. heart beat frequency. The light green, blue, and red rectangles in (f) represent the very–low–frequency (VLF: ∼ 0.05 Hz), low frequency (LF: 0.05–0.15 Hz), and high-frequency (HF: 0.15–0.4 Hz) components, respectively. Experimental data on HRV are not available in the referenced study [1]. The black dotted lines show the results of Model A simulating elderly males whose systemic arterial resistance was set to 0.082 mmHg·s/ml whereas the green dotted lines denote the results of model B simulating adult females whose vital capacity was set to 3.1 L.

20 s after the simulation began. From 20 to 40 s, four parameters—UMAXE = 0.07, TML = 2.0, EXGAIN = 1.0, and PCO2 = 0.02—were applied to simulate active STE.

The predicted results for the BR, TV, MV, HR, and SBP generally aligned with the experimental data for young adult males [1]. However, the predicted DBP was 10–20 mmHg higher than the experimental data. In Fig 1f, the light green, blue, and red rectangles represent the very–low–frequency (VLF: $\sim$ 0.05 Hz), low–frequency (LF: 0.05–0.15 Hz), and high–frequency (HF: 0.15–0.4 Hz) components, respectively, with the VLF component being predominant during active STE. Fig 1 also presents the outcome of two parametric simulations involving Models A and B. In Model A, we simulated elderly males, adjusting their systemic arterial resistance (indicated by $R_i$, a parameter of the circulatory system described in Materials and methods, in Table 2) to 0.082 mmHg·s/ml, while young adult males had a resistance of 0.06 mmHg·s/ml. This choice was informed by the observation that resistance in elderly individuals increases by 37% across the age range from 20 to 80 years old [27]. In Model B, we focused on adult females setting their vital capacity at 3.1 L, while adult males had a vital capacity of 4.8 L, as referenced from Tortora and Derrickson [28]. These simulation results revealed that elderly males (indicated by black dotted lines) exhibited similar changes in HR, SBP, and DBP to the experimental data of elderly males (blue dotted lines). However, they also experienced increased BR, TV, and MV. In contrast, adult female (green dotted lines) demonstrated reduced MV due to decreased TV and lower HR, SBP, and DBP. Notably, the VLF component of HRV in elderly males (black dotted line) and adult females (green dotted line) was lower than that observed in young adult males (red dotted line).

Fig 2 presents a comparative analysis between the simulation results (indicated by red solid lines) and experimental data (represented by blue solid lines) [1]. These data pertain to physiological values versus time during passive STE involving knee flexions. The yellow rectangles highlight the responses observed during the exercise. The physiological parameters considered included the BR, TV, MV, HR, and BP. Fig 2 also presents simulation results (indicated by a red solid line) illustrating the relationship between the heartbeat frequency and the PSD of HRV. Notably, the referenced study [1] did not report experimental data on the HRV. The simulation commenced 20 s after the start of the exercise. During the time interval from 20 to 40 s interval, four parameters—UMAXE = 0.01, TML = 2.0, EXGAIN = 0.001, and PCO2 = 0.02—were utilized to replicate passive STE. The predicted outcome for BR, TV, MV, and SBP closely aligned with the experimental data [1]. While the predicted HR exhibited

**Table 2. Parameters characterizing the circulatory system [66].**

| segment | $C_i$ [ml/mmHg] | $V_i^u$ [ml] | $R_i$ [mmHg·s/ml] | $L_i$ [mmHg·ml/s$^2$] |
|---|---|---|---|---|
| right atrium | 31.5 | 25 | 0.0025 | - |
| right ventricle | - | 40.8 | - | - |
| pulmonary arteries | 0.76 | 0 | 0.023 | $0.18 \times 10^{-3}$ |
| pulmonary peripheral circulation | 5.8 | 123 | 0.0894 | - |
| pulmonary veins | 25.37 | 120 | 0.0056 | - |
| left atrium | 19.23 | 25 | 0.0025 | - |
| left ventricle | - | 16.77 | - | - |
| systemic arteries | 0.28 | 0 | 0.06 | $0.22 \times 10^{-3}$ |
| extrasplanchnic peripheral circulation | 1.67 | 336.6 | 1.407 | - |
| extrasplanchnic venous circulation | 50.0 | 1375 | 0.016 | - |
| splanchnic peripheral circulation | 2.05 | 274.4 | 3.307 | - |
| splanchnic venous circulaiton | 61.11 | 1121 | 0.038 | - |

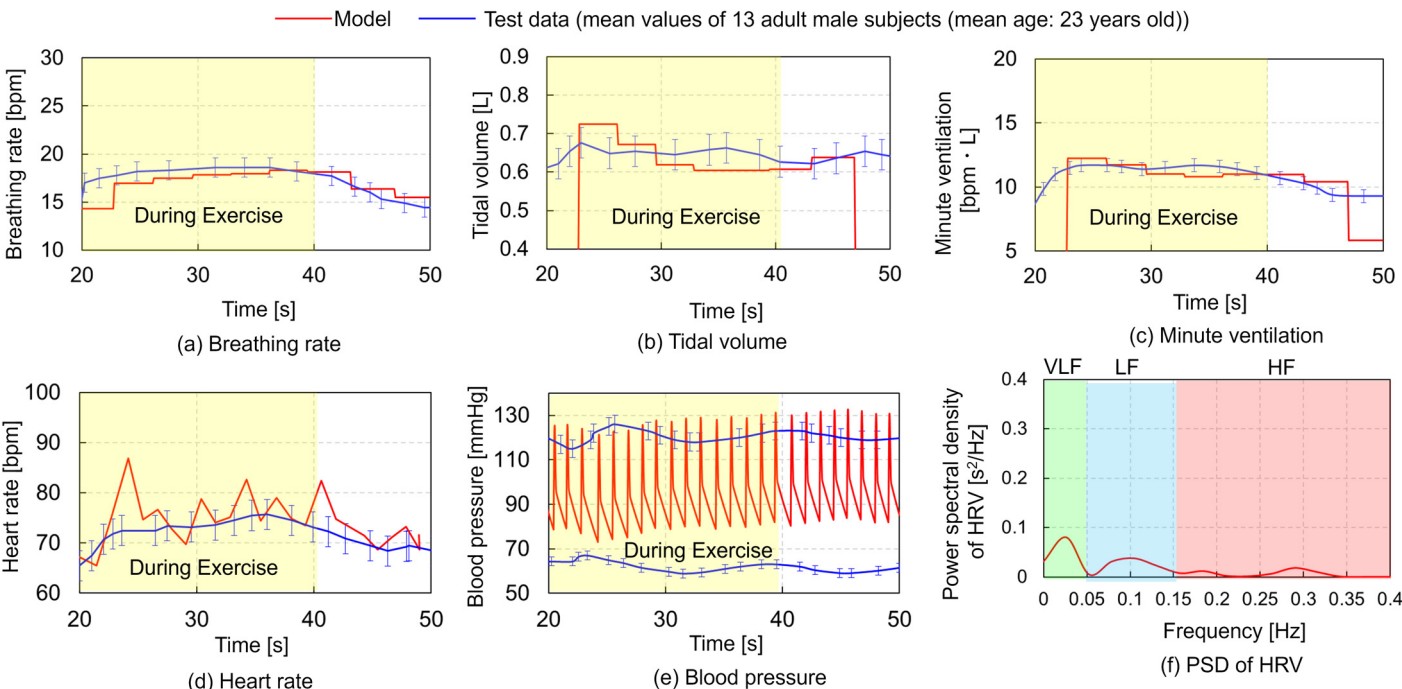

**Fig 2. Comparisons of the simulation results (red solid lines) and experimental data (blue solid lines) [1] with mean values (solid lines) and standard error (error bars) of physiological values vs. time in passive short–term exercise involving knee flexions.** The yellow rectangles represent the responses during the exercise. (a) Breathing rate (BR), (b) Tidal volume (TV), (c) Minute ventilation (MV), (d) Heart rate (HR), (e) Blood pressure (BP) vs. time, and (f) Power spectral density (PSD) of heart rate variability (HRV) vs. heart beat frequency. Experimental data on HRV are not available in the referenced study [1].

peaky waveforms slightly exceeding those in the experimental data, the peak of the predicted DBP was 15 mmHg higher. A comparison between Figs 1f and 2f reveals that the VLF component of HRV during passive STE was one–third lower than that during active STE, while the LF and HF components during passive STE were higher. Simulation results for the first condition indicate that adjusting the parameters of RAMPI, UMAXE, and EXGAIN (see Table 1) can replicate physiological values during active and passive STE.

## Physiological changes during voluntary adjustment of breathing rate

The second condition emulates voluntary BR adjustments, drawing from a prior experimental study [2, 3]. Nuckowska et al. (2019) conducted experiments with eight male and 12 female subjects (mean age: 25.3 years), allowing them to voluntarily control their BR at 6 or 12 beats per minute (bpm) for 5 minutes. Similarly, Song et al. (2003) explored the BR control in five female subjects (mean age: 29 years) across a range of rates (3, 4, 6, 8, 10, 12, and 14 bpm) for 5 minutes. While their study measured HR and BP, they did not report TV data. To maintain a constant MV of 8 bpm·L (as the typical MV for healthy individuals at rest is approximately 8 bpm·L [29]), we adjusted the parameters of PONS, RAMPI, APSR, and UMAX using trial and error to replicate BRs of 6, 8, 10, 12, and 14 bpm and an MV of 8 bpm·L. Fig 3 presents the simulation results for VB controls across five respiratory states A (BR = 14 bpm, TV = 0.57 L), B (BR = 12 bpm, TV = 0.67 L), C (BR = 10 bpm, TV = 0.8 L), D (BR = 8 bpm, TV = 1.0 L), and E (BR = 6 bpm, TV = 1.33 L) under the second condition. The figure encompasses the time histories of BR, TV, MV, HR, and BP, along with the relationship between the PSD of HRV

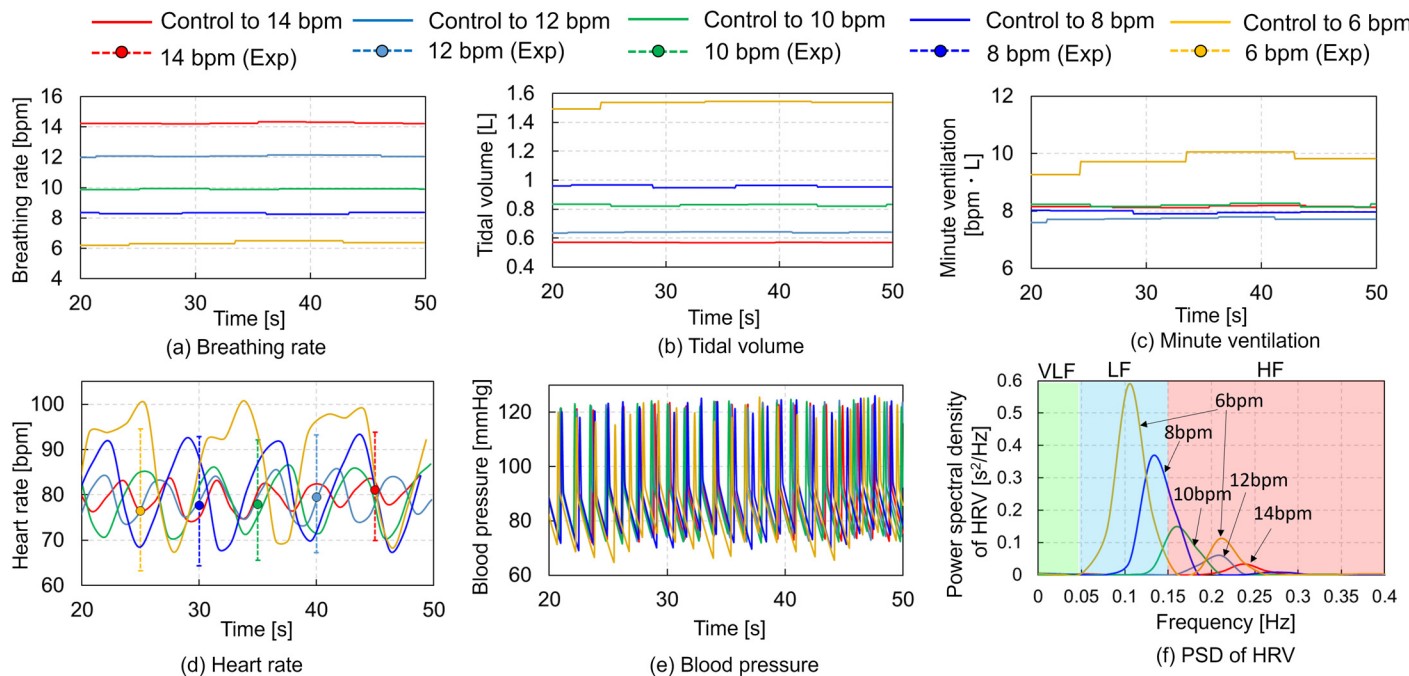

**Fig 3. Simulation results of voluntary breathing controls in five respiratory states: Case A (BR = 14 bpm, TV = 0.5 L), B (BR = 12 bpm, TV = 0.67 L), C (BR = 10 bpm, TV = 0.8 L), D (BR = 8 bpm, TV = 1.0 L), E (BR = 6 bpm, TV = 1.33 L).** (a) Breathing rate (BR), (b) Tidal volume (TV), (c) Minute ventilation (MV), (d) Heart rate (HR) with experimental test data from Song et al. [2], including mean values (solid circles) and peak and trough values (dotted lines), (e) Blood pressure (BP) vs. time, and (f) Power spectral density (PSD) of heart rate variability (HRV) vs. heart beat frequency.

and heartbeat frequency. Fig 3a, 3b and 3c show that the targeted values of BR, TV, and MV were nearly achieved by adjusting the parameters of PONS and RAMPI. Despite efforts to adjust the parameters to achieve an MV of 8 bpm·L in the respiratory state E, TV increased to over 1.4 L, resulting in an MV increase to 10 bpm·L. The variation in HR increased as BR decreased, aligning with the experimental test data from Song et al. [2] (Fig 3d). However, BP remained relatively unchanged (Fig 3e), contrary to Nuckowska et al. [3], who reported a 5 mmHg increase in BP with a decrease in the BR. In the PSD of HRV, the HF component was predominant at BRs of 12 or 14 bpm, while the LF component was predominant at BRs of 6 or 8 bpm. The LF component increased with decreasing BR, consistent with Song et al. [2], who found that LF component is higher at 4 to 8 bpm than at 10 to 14 bpm whereas HF component is higher at 10 to 14 bpm than at 4 to 8 bpm.

## Physiological changes during mental stress loads

The third condition simulates the changes in the HR and BP induced by the MS loads to evaluate the outcomes of an experimental study [15]. Carroll et al. (2000) [15] measured the HR and BP at rest and in response to a 3–minute mental arithmetic stress in 1900 participants using a semi–automatic sphygmomanometer. In our simulations, we were unable to replicate the mental arithmetic stress. Instead, we modeled MS as an electrical input to the amygdala based on prior studies indicating that stress leads to amygdala hyperactivity [30]. We conducted parametric simulations by inputting constant values of 0.1, 0.2, 0.3, 0.4, and 0.5 to the central nucleus of the amygdala (CeA) within the respiratory–circulatory system model, which reproduced automatic breathing at 16 bpm. Fig 4 presents the simulation outcomes of physiological

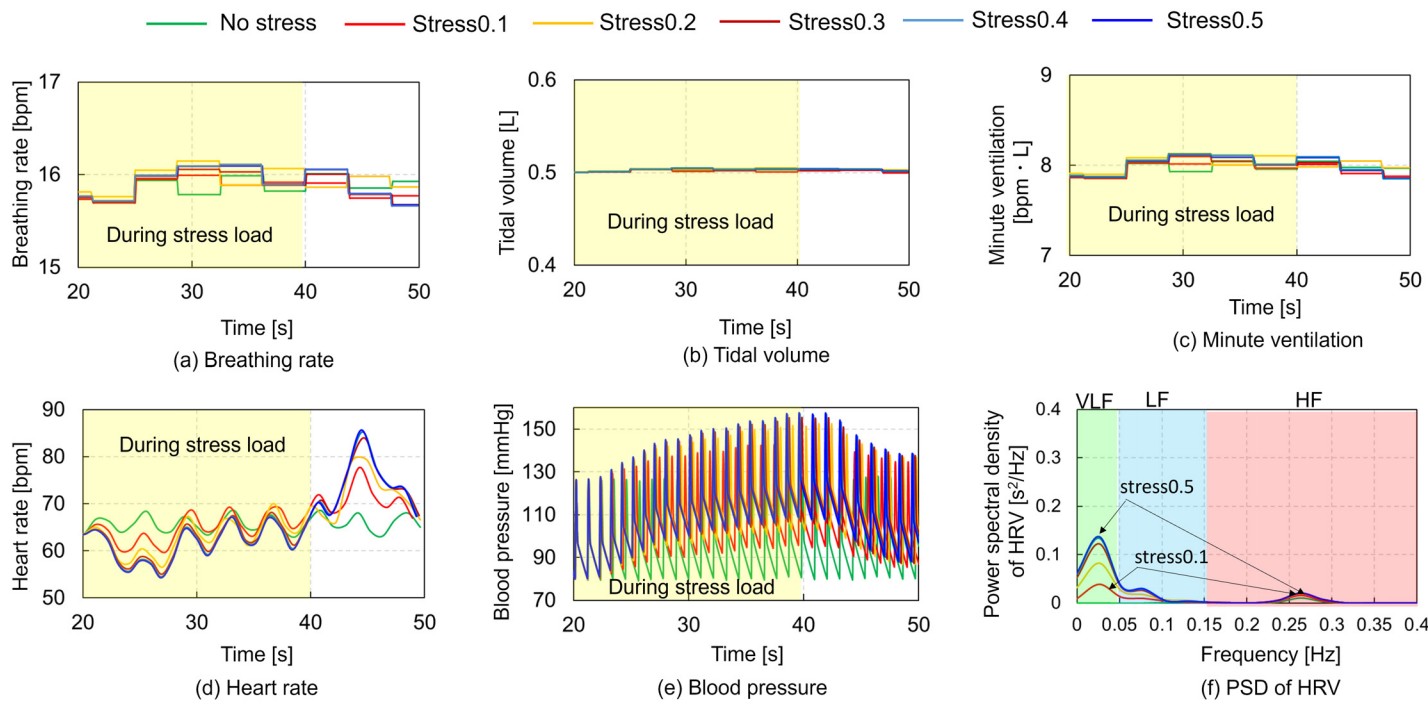

**Fig 4. Simulation results for mental stress loads: (a) Breathing rate (BR), (b) Tidal volume (TV), (c) Minute ventilation (MV), (d) Heart rate (HR), (e) Blood pressure (BP) vs. time, (f) Power spectral density (PSD) of heart rate variability (HRV) vs. heart beat frequency.** The yellow rectangles represent the responses during the mental stress loads. Five mental stress loads varying from 0.1 to 0.5 in steps of 0.1 were applied from 20 to 40 s in the automatic breathing condition (BR = 16 bpm, TV = 0.5 L).

changes under five MS loads, ranging from 0.1 to 0.5 in increments of 0.1, within an automatic breathing condition (BR = 16 bpm, TV = 0.5 L). The figure shows the time histories of BR, TV, MV, HR, and BP, alongside the relationship between the PSD of HRV and heartbeat frequency. Yellow rectangles denote responses to mental stress loads. The predicted respiratory parameters (BR, TV, and MV) remained constant with increasing mental stress load (Fig 4a, 4b and 4c). HR peaked 5 s post–MS load (Fig 4d), while SBP and DBP gradually increased peaking at 40 s, coinciding with the end of the MS load (Fig 4e), This trend aligns with Carroll et al's [15] experimental data, indicating higher HR, SBP, and DBP under the mental arithmetic stress compared to resting state. HRV increased with MS load, with the VLF component being predominant (Fig 4f).

Fig 5a compares the $PaO_2$ levels during active or passive STE, showing higher $PaO_2$ during active STE. Fig 5b compares $PaO_2$ at breathing rates of 6 and 14 bpm, indicating higher $PaO_2$ during slow breathing (6 bpm) compared to rapid breathing (14 bpm). Fig 5c compares $PaO_2$ under MS loads of 0.1 and 0.5, revealing similar $PaO_2$ levels for both stress loads.

Fig 6 presents the HRV metrics: standard deviation of the NN intervals (SDNN), Sample Entropy (SamEn), total HRV power, and LF/HF power ratio across all the simulation conditions. These metrics were derived using the Open–Source Python Toolbox for Heart Rate Variability, pyHRV [31] from RRI time histories. SDNN reflects variability, while SamEn indicates regularity in the RRI time history curve [32]. SDNN was higher in VB controls, in particular, control to slower BR, and lower in passive STE and MS loading (Fig 6a). Conversely, SamEn was higher in passive STE and MS loading and lower in active STE and VB controls (Fig 6b). Total HRV power correlates with cognitive performance [33] and the LF/HF ratio represents

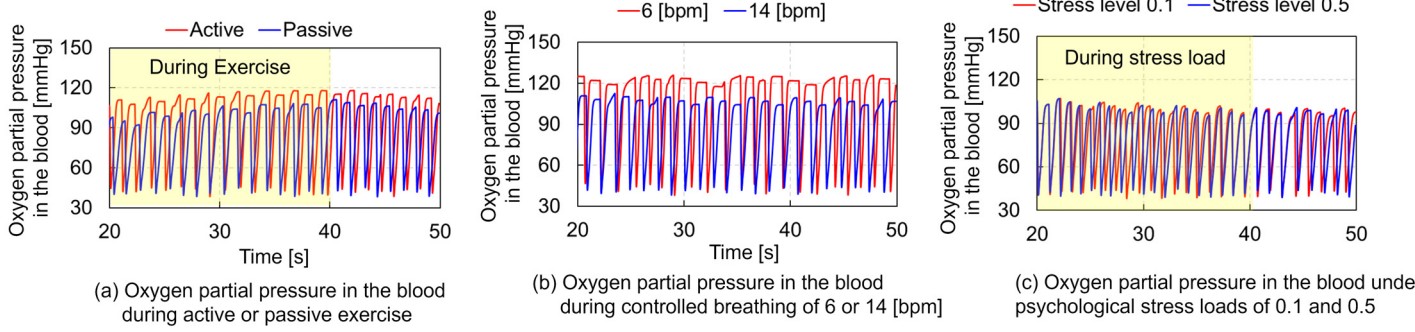

**Fig 5. Comparative analysis of blood oxygen partial pressures during short–term exercise, voluntary controlled breathing, and mental stress loading: (a) active vs passive exercise, (b) controlled breathing at 6 or 14 bpm, and (c) mental stress loads of 0.1 and 0.5.** The yellow rectangles indicate data during short–term exercise and mental stress.

autonomic nervous system balance [32]. The total power of HRV was greater during slower BR and active STE, whereas it was reduced during faster BR and MS loading (Fig 6c). The LF/HF power ratio was higher in active STE and slower BR (Fig 6d). Fig 7 presents Poincaré plots for STE, VB control, and MS loading. The ellipse area of Poincaré plot (EAPP) indicates stress

**Fig 6. HRV metrics of SDNN, Sample Entropy, and total power of HRV and LF/HF power ratio for all simulation conditions.** (a) SDNN, (b) Sample Entropy, (c) Total power of HRV, and (d) LF/HF power ratio.

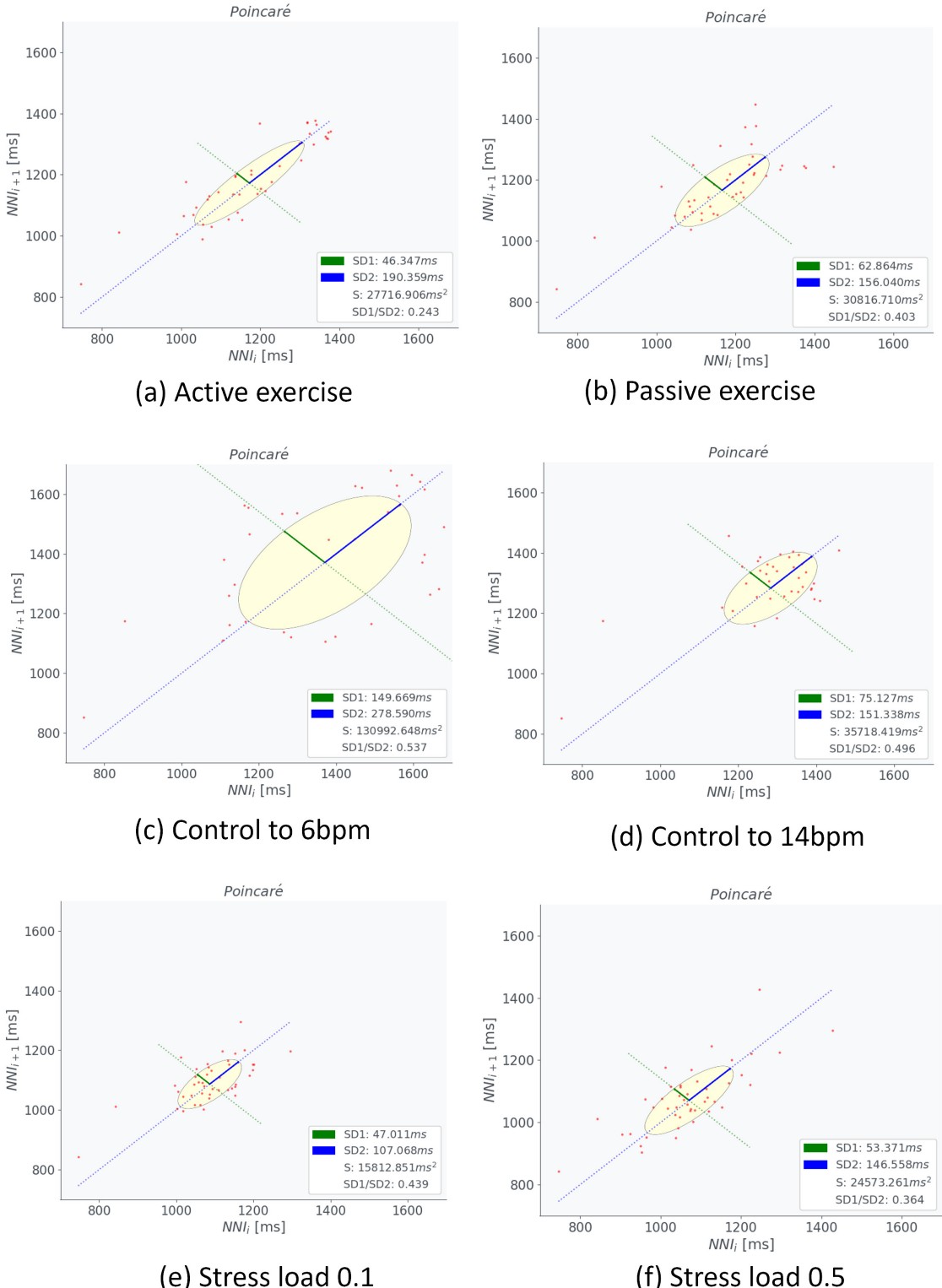

**Fig 7. Comparisons of the Poincaré plots during short–term exercise, voluntary breathing control, and mental stress loading.** (a) active short–term exercise, (b) passive short–term exercise, (c) controlled breathing of 6 bpm, (d) controlled breathing of 14 bpm, (e) mental stress load of 0.1, and (f) mental stress load of 0.5. NNI indicates normal–to–normal interval.

or relaxation states, while the ellipse shape denotes the predominance of sympathetic or para-sympathetic tone [32, 34]. Increased MS loading enhanced sympathetic tone and shortened RR intervals. The EAPP was largest during controlled breathing at 6 bpm, followed by controlled breathing at 14 bpm, passive STE, active STE, MS load of 0.5, and MS load of 0.1.

## Discussion

The computational model developed for predicting the responses of the respiratory and circulatory systems, along with the VB control method using actor–critic reinforcement learning (ACRL) successfully reproduced hyperventilation at the onset of exercise. Figs 1c and 2c show that the model accurately replicated the changes in MV during the STEs exercises, with MV rapidly increasing after 20 s and remaining constant until 40 s in both active and passive STEs. Except for DBP, the predicted BR, TV, MV, HR, and SBP closely matched the experimental data. DBP is primarily influenced by cardiac output and peripheral vascular resistance, which increases and decreases, respectively, during exercises due to the vasodilation of resistance vessels within the exercising skeletal muscles [35]. Therefore, the peripheral systemic resistance $R_{ep}$ in Table 3, one of the regulation effectors for circulatory system described in Materials and methods, must be adjusted to reproduce the DBP. Adjusting $G_4$ of $R_{ep}$ to 0.106 achieved the DBP observed in the experimental data. However, the HR decreased by approximately 10 bpm compared to the experimental data. Further research is required to adjust other parameters to reproduce both HR and DBP simultaneously. The VLF component of HRV was predominant during active STE. However, direct comparison of the predicted HRV with experimental data was impossible due to the absence of HRV data in the referenced study [1]. Several studies have indicated that LF components are elevated during active STE compared to static STE [36], and they increase with higher exercise load from the resting state [37]. Conversely, VLF and LF components increase while HF components decrease in sitting or standing posture with greater muscle activity compared to the supine posture [38, 39]. Additionally, VLF components during rhythmic or random activity are three– to five–fold higher than during rest [40]. PaO$_2$ levels during active STE were higher than during passive STE (Fig 5a), and arterial oxygen partial pressure during STE was higher than at rest [41]. This suggests that our model can reliably predict respiratory and circulatory responses, including HRV and PaO$_2$ during STE. Among HRV metrics, SamEn was higher during passive STE, whereas the LF/HF power ratio was higher during active STE (Fig 6b and 6d). These results indicate that active STE enhances sympathetic nerve activity and reduces regularity in compared to passive STE. Additionally, elderly males with higher systemic arterial resistance than young males replicated HR, SBP, and DBP in experimental data but not BR, TV, and MV. In adult females with lower vital capacity than young males, MV and TV were lower than in young males, reflecting gender differences in ventilatory responses to progressive exercise [42]. The discrepancy in respiratory

**Table 3. Parameters describing the regulation effectors [66].**

| $j$ | $\theta_j$ | $G_j$ | $\tau_j$ [s] | $D_j$ [s] | $\theta_j^0$ |
|---|---|---|---|---|---|
| 1 | $E_{max}^L$ | 0.475 [mmHg/ml·Hz] | 8 | 2 | 2.392 [mmHg/ml] |
| 2 | $E_{max}^R$ | 0.282 [mmHg/ml·Hz] | 8 | 2 | 1.412 [mmHg/ml] |
| 3 | $R_{sp}$ | 0.695 [mmHg·s/ml·Hz] | 6 | 2 | 2.49 [mmHg·s/ml] |
| 4 | $R_{ep}$ | 0.53 [mmHg·s/ml·Hz] | 6 | 2 | 0.78 [mmHg·s/ml] |
| 5 | $V_{sv}^u$ | -265.4 [ml/Hz] | 20 | 5 | 1435.4 [ml] |
| 6 | $V_{ev}^u$ | -132.5 [ml/Hz] | 20 | 5 | 1537 [ml] |
| 7 | $T$ | -0.13 [s/Hz] | 2 | 2 | 0.58 [s] |

states for elderly males likely arises from difference in ribcage geometry and stiffness between young and elderly adults. Further research is required to examine the impact of age–related changes in ribcage geometry and stiffness on respiratory function.

The model successfully replicated various respiratory states, from rapid to slow breathing. The maximum heart rate was higher during slow breathing compared to rapid breathing (Fig 3d). Song et al. [2] examined the peak, mean, and trough HR values for BR ranging from 3 to 14 bpm. Comparing the results for 14 bpm with those for 6 bpm as reported by Song et al. [2], the mean and trough values for 6 bpm were lower by 5 and 6 bpm, respectively. The predictions of the model demonstrated that the trough values at 6 bpm were 6 bpm lower than those at 14 bpm, consistent with the experimental data [2], although the mean values at 6 bpm were 5 bpm higher than those at 14 bpm, differing from the experimental data [2]. In the simulation results, the peak and mean values were higher during slow breathing compared to rapid breathing. However, the HR controlled at 6 bpm varied significantly from 100 bpm to 70 bpm within less than 5 s, replicating the cardiac oscillations during deep slow breathing at 6 bpm as reported by Sevoz–Couche and Laborde [43]. No predicted changes in SBP and DBP were observed in any respiratory state. Contradictory results exist regarding changes in SBP and DBP with increasing BR. SBP and DBP decreased by approximately 3 mmHg with an increase in BR from 6 bpm to 12 bpm [3], whereas they increased by 2 to 4 mmHg with an increase in BR from 6 bpm to 15 bpm [44]. The model predicted an increase in LF components with a decrease in breathing speed (Fig 3f). Experimental data corroborated this, showing increased LF components during slow breathing [2, 45]. $PaO_2$ levels were higher during slow breathing (6 bpm) compared to rapid breathing (14 bpm) (Fig 5b). Slow breathing enhances HRV [2] and $PaO_2$ [46]. Among HRV metrics, SDNN, total HRV power, and LF/HF power ratio were higher during slow breathing (6 and 8 bpm) than during rapid breathing (10–14 bpm) (Fig 6a, 6c and 6d). EAPP was highest during slow breathing at 6 bpm, followed by rapid breathing at 14 bpm across all simulation conditions (Fig 7). These findings indicate that our model reliably predicts respiratory and circulatory responses, including HRV and PaO2, during VB. Slow breathing enhances variability, cognitive performance, and relaxation.

The model predicted increases in HR and SBP for MS loads (Fig 4d and 4e), consistent with previous experimental studies [8, 15, 47, 48]. The predicted respiratory states–BR, TV, and MV–remained stable across all stress conditions, despite reports of hyperventilation in response to stress [49] and increased BR due to anxiety anticipation [16]. The predicted HRV, particularly the VLF components, increased with an increase in the MS load. Experimental studies showed that HRV variables changed in response to stress due to low parasympathetic activity, characterized by a decrease in the HF band and an increase in the LF band [50], aligning with our predictions. However, differences were observed in the VLF components, which decreased with increasing depression [51] and contributed to MS recovery [52]. The peak VLF component predicted by the model was approximately 0.15 $s^2$/Hz, less than half of those during active STE and voluntary slow, deep breathing. The $PaO_2$ under an MS load of 0.5 was almost similar to that under an MS load of 0.1 (Fig 5c). All HRV metrics and the EAPP were higher under an MS load of 0.5 than under an MS load of 0.1 (Figs 6 and 7). SamEn was highest under an MS load of 0.5 among all simulation conditions, and the total power of HRV was lower under MS loading than during active STE and slow breathing. The EAPP under MS loads of 0.1 and 0.5 was significantly lower than during slow breathing at 6 bpm. These findings indicate that our model can reliably predict HR and BP under MS loading and that MS loading enhances regularity while reducing relaxation levels. However, the current model does not account for the amygdala–LC interaction, where fear– and stress–induced activity in the amygdala affects sensory brain regions via LC connections [53] and the entrained LC neurons via respiration [25, 54], nor the amygdala–prefrontal cortex

(PFC)–hippocampus (HPC) interaction, which may relate to anxiety anticipation [55]. Further research is required to explore the amygdala–LC and amygdala–PFC–HPC interactions, examine the relationship between respiration and MS, and investigate the VLF components' relation to MS loads.

These simulation results demonstrated that voluntary slow deep breathing at 6 bpm increased the $PaO_2$, total HRV power, SDNN, and EAPP, thereby enhancing the cognitive performance, variability, and relaxation. Conversely, MS loading elevated HR, SBP, and SamEn while reducing total HRV power and EAPP, leading to increased physical risk and decreased relaxation. Individuals with high resting HRV outperformed those with low HRV, exhibiting faster reaction times in stimulus identification [33]. Exercise and meditation with controlled VB can elevate resting HRV level. Lower HRV is associated with increases SBP, hypertension and cardiovascular disease, as shown in a longitudinal study on rheumatoid arthritis patients [56]. Both active STE [41] and slow breathing [46] elevate $PaO_2$. Thus, slow breathing promotes mental health whereas MS loading increases the physical risk and reduces relaxation.

While our proposed model does not predict emotional arousal or valence levels differently from existing literature [21–26], it uniquely predicts temporal changes in physiological values such as BR, TV, MV, HR, SBP, DBP, HRV, and blood oxygen and carbon dioxide partial pressures. It elucidates the mutual interactions among these physiological values during resting and active states, including active STE and VB control. This capability may enhance the prediction accuracy of emotional arousal or valence in AI–based data analysis using large datasets of physiological or psychological measurement. Additionally, it provides insight into the relationship between (1) the activity of emotion–related brain regions (e.g., amygdala, anterior cingulate cortex, and PFC) modeled by spiking neuron models, and (2) physiological values such as HR and HRV, which have been experimentally verified [57–61]. Furthermore, the model parameters for systemic arterial resistance and vital capacity reproduce HR and BP in age–related differences and MV and TV in gender–related difference. Thus, our approach has the potential to elucidate the mutual interactions among physiological values and investigate the effect of individual differences during VB, STE, or MS.

This study has several limitations. First, the model only includes the diaphragm and abdominal muscles, focusing solely on abdominal breathing. Normal breathing involves both abdominal and chest breathing, necessitating the inclusion of other respiratory muscles such as the external intercostal muscles, inspiratory accessory muscles (scalene and sternocleido-mastoid), and expiratory muscles (internal intercostal muscles) to simulate chest breathing accurately. Second, the current model cannot adjust the inspiratory–to–expiratory ratio and only simulates mouth breathing, thus precluding the investigation of the effects of the inspiratory–to–expiratory ratio on nasal breathing during slow deep breathing. Nasal breathing reduces mean blood pressure and DBP but does not affect SBP or HR while increasing parasympathetic contributions to HRV [62]. Further research is needed to model nasal breathing, adjust the inspiratory–to–expiratory ratio during slow deep breathing, and investigate the effects of slow, deep breathing on the autonomic nervous system and MS reduction. Third, the current model can predict respiratory and circulatory responses during long–term exercise exceeding four minutes; however, it predicts MV fluctuations with an amplitude of 1 bpm, which is not observed in experimental data. Additional studies are required to minimize these fluctuations for accurate long–term exercise simulation. Lastly, only four parameters were used as inputs to the respiratory–circulatory system model to represent active and passive STE. Future research should integrate this model with a musculoskeletal model, including muscle controllers [63], to simulate physiological values such as BR, HR, BP, and HRV during various motions and posture stabilizations.

## Materials and methods

### Integrated respiratory and circulatory systems

Fig 8 shows a computational model for predicting the responses of the respiratory and circulatory systems. The model comprises the nucleus of the solitary tract (NTS), respiratory center and system as rhythm–generation and pattern formation regions, and circulatory center and system. The respiratory center and system were developed based on a mathematical model of the closed–loop control of breathing proposed by Molkov et al. [64]. Lung ventilation, or breathing, involves the exchange of air between the lungs and the environment, driven by the rhythmic contraction of the diaphragm and abdominal muscles. The firing activities of the phrenic and abdominal motor neurons, which control the diaphragm and abdominal muscles, respectively, are the primary outputs of the brainstem respiratory central pattern generator (CPG) that generates respiratory oscillations. The closed–loop respiratory system comprises two primary feedback pathways from the lungs to the respiratory CPG: mechanical and chemical feedback. Mechanical feedback is mediated by pulmonary stretch receptors (PSR) in the

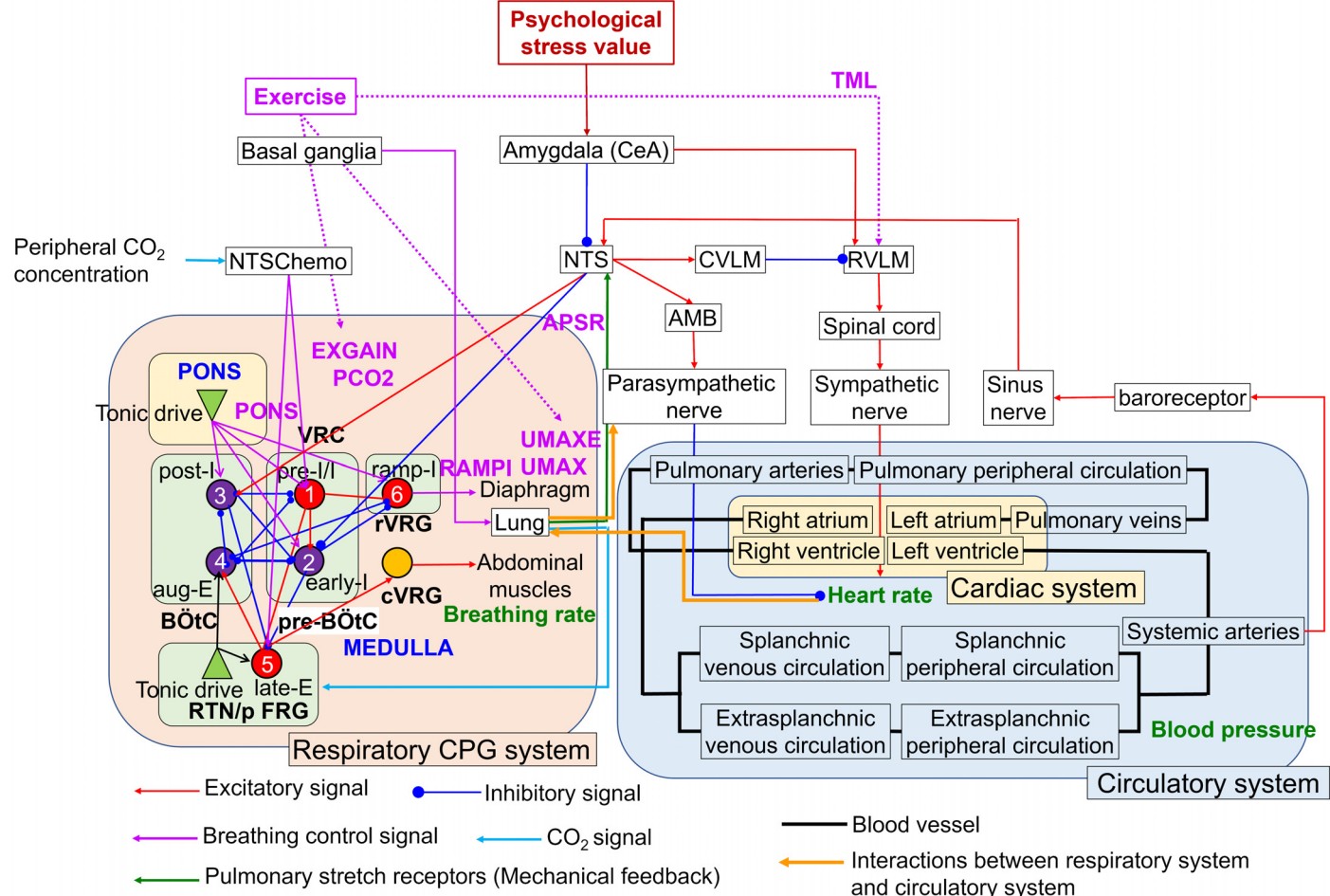

**Fig 8. Respiratory–circulatory system model.** The model integrates the respiratory CPG system with spiking neurons, as proposed by Molkov et al. (2014) [64], and the circulatory system utilizing a rate–coding approach, as described by Ursino (1998) [66]. The red, blue, light blue, and purple lines denote excitatory and inhibitory signals, $CO_2$ signals as chemical feedback, and breathing control signals, respectively. The light green line indicates pulmonary stretch receptors as mechanical feedback. Thick black and orange lines represent blood vessels and the interactions between the respiratory and circulatory systems, respectively.

lungs, which convey lung volume information to the brainstem via the vagus nerve. Chemical feedback is provided by central chemoreceptors in the retrotrapezoid nucleus (RTN) neurons, which are highly sensitive to oxygen ($O_2$) and carbon dioxide ($CO_2$) levels in brain tissues. Additionally, we incorporated chemical feedback from second–order peripheral chemoreceptors, which are sensitive to $O_2$ and $CO_2$ levels in the blood and project directly the RTN and pre–I/I (pre–BötC) via chemoreceptors in the NTS (NTS Chemo) as described by Barnett et al. [65], thereby modulating the respiratory CPG in a $CO_2$–dependent manner.

In the respiratory center and system (Fig 8), each neuron in the respiratory CPG can be fundamentally modeled using Hodgkin–Huxley (HH) neurons. The model is represented by the following equations:

$$C\dot{V}_i = -g_L(V_i - E_L) - g_{Kdr}n_K^4(V_i - E_K) - I_i - s_i(V_i - E_E) - q_i(V_i - E_I), \tag{1}$$

$$n_K = 1.0/(1.0 + \exp(-(V_i + 30.0)/4.0))^4, \tag{2}$$

where $C$ is the membrane capacitance ($C$ = 20 pF), and $g_K$ and $g_L$ are the peak conductance of potassium and leak conductance (leak), respectively. $g_{Kdr}$ = 5 nS and $g_L$ = 2.8 nS, where $E_K$ and $E_L$ represent the reversal potential of potassium and leakage reverse potential, respectively. Specifically, $E_K$ = –85 mV and $E_L$ = –75 mV. The currents $I_i$ represents the Pre–I/I (i = 1), Early–I (i = 2), Post–I (i = 3), Aug–E (i = 4), Late–E (i = 5), and Ramp–I (i = 6), and are defined as follows:

$$I_i = \begin{cases} g_{NaP}m_{Na}h_i(V_i - E_{Na}) & (i = 1, 5) \\ g_{AD}m_i(V_i - E_K) & (i = 2, 3, 4) \\ 0 & (i = 6) \end{cases} \tag{3}$$

where $g_{NaP}$ and $g_{AD}$ are the maximum conductances, and $E_{Na}$ and $E_K$ are the reversal potentials for sodium and potassium, respectively. The values are $g_{NaP}$ = 5 nS, $g_{AD}$ = 10 nS, and $E_{Na}$ = 50 mV.

The gating parameters $m_{Na}$, $m_i(i = 2, 3, 4)$, and $h_i(i = 1, 5)$ are functions of the membrane variable $V_i$ and are defined by the following equations:

$$m_{Na} = 1.0/(1.0 + \exp(-(V_i + 40.0)/6.0)), \tag{4}$$

$$h_{Na}^\infty = 1.0/(1.0 + \exp((V_i + 55.0)/10.0)), \tag{5}$$

$$\tau_h \dot{h}_i = h_{Na}^\infty - h_i, \tag{6}$$

$$\tau_{m_i} \dot{m}_i = \tilde{V}_i - m_i, \tag{7}$$

where $\tau_h$ = 4.0/cosh(($V_i$ + 55.0)/10.0) s and $\tau_{m_i}$ (i = 2,3,4) are time constants. The variable $\tilde{V}_i$ represents the neuronal activation level and is defined as follows:

$$\tilde{V}_i = \begin{cases} 0 & (V_i < -50 \text{ [mV]}) \\ \left(\frac{V_i+50}{30}\right)^k & (-50 \text{ [mV]} \leq V_i \leq -20 \text{ [mV]}) \\ 1 & (V_i > -20 \text{ [mV]}). \end{cases} \tag{8}$$

The neuronal activation levels of late–E (i = 5) and ramp–I (i = 6) were utilized to simulate the abdominal muscles and diaphragm, respectively. This study introduced the power exponent $k$

to regulate the TV, corresponding to the breathing control parameter RAMPI described in the Results section; $k = 1$ for $i = 1 \sim 5$ and $k = $ RAMPI for $i = 6$.

The gating variables $s_i$ and $q_i$ for synaptic conductances were derived from the activity of the presynaptic neurons $\tilde{V}_i$ and other input sources using the following equations:

$$s_i = \sum_{j=1}^{5} a_{ji} \cdot \tilde{V}_j + \sum_{k=1}^{2} c_{ki} \cdot D_k + e_i \cdot \text{PSR} + g_i \cdot \text{EXGAIN} \cdot (p_c + \text{PCO2}), \tag{9}$$

$$q_i = \sum_{j=1}^{5} b_{ji} \cdot \tilde{V}_j + f_i \cdot \text{PSR}, \tag{10}$$

$$D_2 = \tanh(P_S/P_{S0} + \Delta P_c), \tag{11}$$

$$\tau_S \cdot \dot{P}_S = p_{ce} - P_S, \tag{12}$$

$$\dot{p}_c = \frac{D_c}{\sigma_c \cdot V_c \cdot C_u} \cdot (P_{ac} - P_c) + \delta \cdot \left( \frac{l_2 \cdot z \cdot H}{\sigma_c} - r_2 \cdot p_c \right), \tag{13}$$

where $\tilde{V}_i$ is defined in Eq (8). The constant drive $D_1$ originates from the pons and corresponds to the breathing control parameter PONS described in the Results section. Drive $D_2$ is related to the partial pressure of $CO_2$ in the blood that projects to each neuron in the respiratory CPG from the RTN (see [65] for more detailed information). The synaptic weights ($a, b, c, e, f$) were modified only for $c_{1i}$ from the original model [65] and are listed in Table 4. Eqs (1), (6), (7) and (12) were updated using the Euler method. In this study, the time step was set to 0.0001s.

For VB control, we adapted the diaphragm and abdominal muscles models using a Hill–type muscle model as described by O'Connor et al. [67]. We applied Eqs (14) and (15) to calculate the recoil pressures of the diaphragm and abdominal wall.

$$\sigma_{di} = u_{di}\sigma_{di}^{max}F_{fl}^{di}F_{fv}^{di} + \sigma_{di}^{psv} + R_{di}\dot{V}_{di}, \tag{14}$$

where $\sigma_{di}^{max}$ is the static diaphragm recoil pressure at optimal length and maximum activation; $u_{di}$ is the phrenic activation of the diaphragm; $R_{di}\dot{V}_{di}$ represents the pressure due to the passive resistance of the diaphragm; $F_{fl}^{di}$ is the static pressure–volume relationship of the diaphragm; and $F_{fv}^{di}$ is the pressure–flow relationship of the diaphragm with velocity replaced by flow. $\sigma_{di}^{psv}$ is the passive transdiaphragmatic pressure as a function of the diaphragm volume. Detailed descriptions of $F_{fl}^{di}$, $F_{fv}^{di}$, and $\sigma_{di}^{psv}$ are provided by O'Connor et al. [67].

$$\sigma_{ab} = u_{ab}F_{CEmax}F_{fl}^{ab}F_{fv}^{ab}\left(\frac{k_{ab}}{r_t} + \frac{k_{ab}}{r_s}\right) + \frac{V_{ab} - V_{ab0}}{C_{ab}} + R_{ab}\dot{V}_{ab}, \tag{15}$$

**Table 4. Parameters of respiratory CPG model.**

|  | Pre–I/I | Early–I | Post–I | Aug–E | Late–E | Pons | RTN | PSR |
|---|---|---|---|---|---|---|---|---|
| i | $a_{1i}$ | $b_{2i}$ | $b_{3i}$ | $b_{4i}$ | $a_{5i}$ | $c_{1i}$ | $c_{2i}$ | $e_i, f_i$ |
| 1: Pre–I/I | 0 | 0 | 1.0 | 0.15 | 0.5 | 0.6 | 0.2 | |
| 2: Early–I | 0.35 | 0 | 0.42 | 0.15 | 0 | 0.6 | 0.5 | $f_2 = 4$ |
| 3: Post–I | 0 | 0.22 | 0 | 0 | 0 | 1.05 | 0 | $e_4 = 20$ |
| 4: Aug–E | 0 | 0.42 | 0.2 | 0 | 0.25 | 0 | 1.0 | |
| 5: Late–E | 0 | 0.075 | 0.12 | 0 | 0 | 0 | 0.15 | |
| 6: Ramp–I | 0.35 | 0.3 | 0.7 | 0.7 | 0 | 0.25 | 1.1 | |

where $u_{ab}$ represents the activation of the abdominal muscle, and $F_{CEmax}$ is the maximal force capacity of the contractile element for a 1.5 cm$^2$ cross–section of the canine external oblique muscle. $F_{fl}^{ab}$ is the static force–length relationship of the abdominal wall, and $F_{fv}^{ab}$ is the force–velocity relationship of the abdominal wall muscle. $R_{ab}\dot{V}_{ab}$ accounts for the pressure due to the passive resistance of the abdominal wall muscles. The constant $k_{ab}$ converts force to surface tension, while $(1/r_s + 1/r_t)$ translates the surface tension to pressure via Laplace's law. The term $(V_{ab} - V_{ab0})/C_{ab}$ represents the passive recoil pressure of the abdominal wall, where $V_{ab0}$ is the volume at which the recoil pressure is zero, and $C_{ab}$ denotes the compliance of the abdominal wall.

The respiratory CPG model determines the lung volume $V_L$ by solving Eq (16) for the pressure equilibrium involving abdominal pressure $\sigma_{ab}$, pleural pressure $P_{pl}$, and diaphragm recoil pressure $\sigma_{di}$.

$$\sigma_{ab} - P_{pl} = \sigma_{di}, \tag{16}$$

where, $P_{pl}$ is related to lung volume $V_L$ through Eq (17).

$$P_{pl} = -R_{rs}\dot{V}_L - (V_L - V_{L0})/C_L, \tag{17}$$

where $R_{rs}$ is the airway resistance, $V_{L0}$ is the lung volume at zero recoil pressure, and $C_L$ is the lung compliance.

In addition, we modified several equations from the original model by O'Connor et al. [67] as follows:

$$V_{L0} = V_L^{VRC}, \tag{18}$$

$$\sigma_{di}^{TLC} = \sigma_L^{TLC} + \sigma_{ab}^T, \tag{19}$$

$$\begin{cases} V_{rc}^{kmFRC} + V_{ab}^{kmFRC} = (V_L^{FRC} - V_L^{RV})/\text{VC} \\ V_{rc}^{kmFRC} : V_{ab}k^{kmFRC} = 0.1282 : 0.04, \end{cases} \tag{20}$$

where the vital capacity (VC), functional residual capacity (FRC) $V_L^{FRC}$, and residual volume (RV) $V_L^{RV}$ are set to 4.8, 2.2, and 1.2 L, respectively. To adapt the original adult rat respiratory system model [64] to the adult human respiratory system, we adjusted the parameters listed in Table 5 and revised the equation for the activity of the PSR (ref. Eq. (25) in [64]) as follows:

$$PSR = 0.0005 \cdot cPSR \cdot (V_L - V_L^{RV})^2, \tag{21}$$

where $V_L^{RV}$ is the residual volume of the lung. cPSR is a coefficient that adjusts the effect of the

**Table 5. Parameters adjusted for adult human respiratory system.**

| | Original adult rat [64] | Modified adult human [67, 68] |
|---|---:|---:|
| Lung elastance E [mmHg/L] | 3,000 | 3.659 |
| Volume of the lung capillaries $V_c$ [L] | 0.000192 | 0.070 |
| Acceleration rate of the chemical reaction $\delta$ [-] | 500 | $10^{1.9}$ |
| Diffusion capacity of $CO_2$ $D_C$ [L/mmHg/s] | $10^{-4}$ | $7.08 \times 10^{-3}$ |
| Diffusion capacity of $O_2$ $D_O$ [L/mmHg/s] | $5 \times 10^{-6}$ | $3.5 \times 10^{-4}$ |
| Time constant of gating variable m2 $\tau_{m2}$ [s] | 0.67 | 2.0 |
| Time constant of gating variable m3 $\tau_{m3}$ [s] | 0.67 | 2.0 |

PSR on the lung volume change, corresponding to the breathing control parameter APSR described in the Results section. We set $V_L^{RV} = 1.387L$.

The circulatory center was developed based on the pathways by which the CeA influences blood pressure during mental stress or anxiety, as proposed by [69], and the autonomic chronotropic control of the heart, as described by [70]. During stress, the CeA may inhibit the baroreceptive neurons in the NTS, potentially deactivating inhibitory inputs from the caudal ventrolateral medulla (CVLM) to the rostral ventrolateral medulla (RVLM). This could activate RVLM neurons, increasing sympathetic outflow, BP, and HR. The nucleus ambiguus (AMB) also receives excitatory signals from the NTS, enhancing parasympathetic outflow and reducing HR. Baroreceptors in the carotid arteries and aortic arch detect increased in the arterial blood pressure, activating afferent nerves (sinus nerves) and NTS neurons. The circulatory center and system were modeled using a rate–coding approach based on Ursino (1998) [66] with parameters for each segment in the circulatory system (Fig 8), listed in Table 2.

In the circulatory center and system (Fig 8), the afferent baroreflex pathway is modeled as a first–order linear partial differential equation (Eq 22) using the carotid sinus pressure $P_{cs}$. The frequency of spikes in the afferent fibers, $f_{cs}$, is represented as a sigmoidal function (Eq 23) with an intermediate variable $\tilde{P}$.

$$\tau_p \frac{d\tilde{P}}{dt} = P_{cs} + \tau_z \frac{dP_{cs}}{dt} - \tilde{P}, \tag{22}$$

$$f_{cs} = f_{min} + f_{max} \cdot \exp\left(\frac{\tilde{P} - P_n}{K_a}\right) / \left(1 + \exp\left(\frac{\tilde{P} - P_n}{K_a}\right)\right), \tag{23}$$

where $\tau_p$ and $\tau_z$ are time constants $\tau_p$ = 2.076 s and $\tau_z$ = 6.37 s. $f_{max}$ and $f_{min}$ are the upper and lower saturations limits of the frequency discharge, respectively; $f_{min}$ = 2.52 spikes/s, $f_{max}$ = 47.78 spikes/s, $P_n$ is the intrasinus pressure at the central point of the sigmoidal function, and $K_a$ is a parameter with pressure dimensions $P_n$ = 92.0 mmHg and $K_a$ = 11.758 mmHg.

The frequencies of spikes in the efferent sympathetic nerves, $f_{es}$, and the efferent vagal fibers, $f_{ev}$, are described by the frequency of spikes in the afferent fibers, $f_{cs}$, as follows:

$$f_{es} = f_{es_{inf}} + (f_{es_0} - f_{es_{inf}}) \cdot \exp(-k_{es} \cdot f_{cs}) + \text{TML}, \tag{24}$$

$$
\begin{aligned}
f_{ev} = {} & f_{ev_0} + f_{ev_{inf}} \cdot \exp\left(\frac{f_{cs} - f_{cs_0}}{k_{ev}} - k_{resp} \cdot (V_L - V_{L0})\right) \\
& / \left(1 + \exp\left(\frac{f_{cs} - f_{cs_0}}{k_{ev}} - k_{resp} \cdot (V_L - V_{L0})\right)\right),
\end{aligned} \tag{25}
$$

where $f_{es_{inf}}$, $f_{es_0}$, and $k_{es}$ are constants with values $f_{es_{inf}}$ = 2.1 spikes/s, $f_{es_0}$ = 16.11 spikes/s, and $k_{es}$ = 0.0675 s. Similarly, $f_{ev_{inf}}$, $f_{ev_0}$, $f_{cs_0}$, $k_{ev}$, and $k_{resp}$ are constants with values $f_{ev_{inf}}$ = 9 spikes/s, $f_{ev_0}$ = 5.5 spikes/s, $f_{cs_0}$ = 25.0 spikes/s, $k_{ev}$ = 7.06 spikes/s, and $k_{resp}$ = 0.08 × $10^{-3}$ $m^3$.

As the state variables of the circulatory system are changed by efferent sympathetic nerve stimulation, the controlled variables $\theta_j$, are defined as follows: $E_{max}^L$ and $E_{max}^R$, represent the end–systolic elastances of the left and right ventricles ($j$ = 1, 2); $R_{sp}$ and $R_{ep}$ denote the hydraulic resistances of the splanchnic and extrasplanchnic peripheral circulations ($j$ = 3, 4), respectively; $V_{sv}^u$ and $V_{ev}^u$ indicate the unstressed volumes of the splanchnic and extrasplanchnic venous circulation ($j$ = 5, 6), respectively; and $T$ signifies the heart period ($j$ = 7). The parameters of $\theta_j$ are detailed in Table 3 [29]. The responses of resistances, unstressed volumes, and cardiac elastances to sympathetic drive encompass pure latency, a monotonic logarithmic

static function, and low–pass first–order dynamics. Consequently, the following equations apply:

$$\overline{\Delta\theta_j(t)} = \begin{cases} G_j \ln[f_{es}(t - D_j) - f_{esmin} + 1] & \text{if } f_{es}(t - D_j) \geq f_{esmin}, \\ 0 & \text{if } f_{es}(t - D_j) < f_{esmin}, \end{cases} \tag{26}$$

$$\tau_j \frac{d\Delta\theta_j(t)}{dt} = -\Delta\theta_j(t) + \overline{\Delta\theta_j(t)}. \tag{27}$$

Here, $\overline{\Delta\theta_j}$ is an intermediate variable with static characteristics, and $G_j$ represents the constant gain factor of each component. The parameters $\tau_j$ and $D_j$ denote the time constant and pure latency, respectively. $f_{esmin}$ is the minimum sympathetic stimulation and $f_{esmin}$ = 2.66 Hz according to [66]. The change in heart period due to efferent vagus nerve stimulation is given by:

$$\overline{\Delta T_v}(t) = G_v f_{ev}(t - D_v), \tag{28}$$

$$\tau_v \frac{d\Delta T_v(t)}{dt} = -\Delta T_v(t) + \overline{\Delta T_v}(t). \tag{29}$$

$\overline{\Delta T_v}$ is an intermediate variable that exhibits static characteristics. $G_v$ = 0.09 s/Hz, $D_v$ = 0.2 s, $\tau_v$ = 1.5 s.

$$\theta_j(t) = \begin{cases} \Delta\theta_j(t) + \theta_j^0 & (j = 1 - 6), \\ \Delta T_v + \Delta\theta_j(t) + \theta_j^0 & (j = 7). \end{cases} \tag{30}$$

In Eq (30), $\Delta T_v$ and $\Delta\theta_7$ denote the variations in the heart period regulated by the sympathetic and parasympathetic nerves, respectively. The parameter $\theta_7^0$ represents the baseline heart period without efferent nerves stimulation, set to 0.58 s.

Although the partial pressures of $CO_2$ and $O_2$ in the blood are reinitialized with each heartbeat in the model by Molkov et al. [64], the heartbeat initiation time was derived from changes in heart period due to autonomic nerve activity described in Eq (30) for $j$ = 7. The lung volume $V_L$ from the respiratory system influenced parasympathetic nerve activity, as detailed in Eq (26). The total pressure and concentration of $O_2$ and $CO_2$ in the mouth were set to 760 [mmHg], 21%, and 0.03%, respectively. A comprehensive description of the circulatory center and system is available in our previous study [71].

## Voluntary breathing control method using reinforcement learning

Fig 9 shows the VB control model. Anatomical connections between the substantia nigra in the basal ganglia and the respiratory control centers, including a direct pathway to the pre–Bötzinger complex, are illustrated. The output of the substantia nigra indirectly informs the respiratory control centers about other ongoing movements [72, 73]. Consequently, we determined the voluntary activation levels of the diaphragm and abdominal muscles using ACRL, a mathematical model of the basal ganglia, based on [64]. In contrast, the involuntary activation levels of these muscles were derived from previously described respiratory and circulatory system models.

The VB control model (Fig 9) was developed based on a previous study by the authors [63]. Critic and actor networks were implemented using a normalized Gaussian network (NGnet)

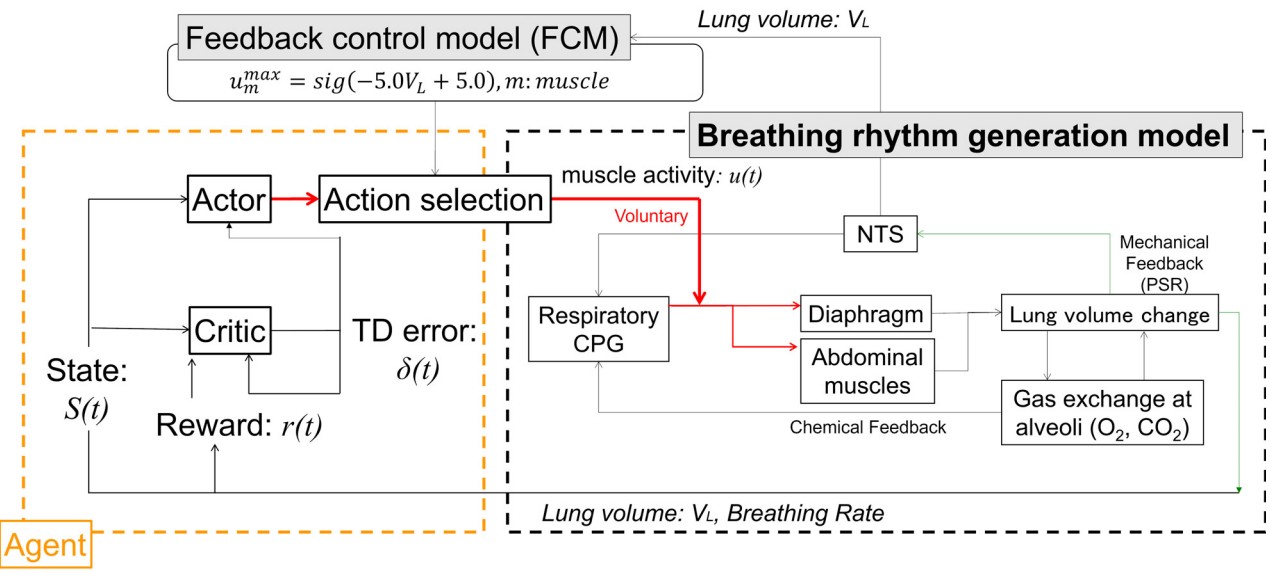

**Fig 9. Voluntary breathing control model.** The model incorporates a muscle controller utilizing actor–critic reinforcement learning to simulate the activity of respiratory muscles (diaphragm and abdominal muscles) during voluntary breathing. Additionally, it includes a breathing rhythm generation model analogous to the respiratory CPG system. The red lines denote muscle activation signals for the respiratory muscles.

and a continuous time–space formulation for reinforcement learning [74]. NGnet models continuous state space using a Gaussian softmax network, which can generalize the state space by extrapolation, even outside the range, as a base function of the radial basis function network. Two state spaces were set: the first, $s_1$, is the difference between the current partial pressure of $CO_2$ in the blood ($p_c$ in Eq (9)) and the mean of the maximum value (45 mmHg) and minimum value (35 mmHg) of $p_c$. The second, $s_2$, is the difference between the percentage of $CO_2$ in the air inside the mouth and the reference value, set to 0.03% in this study. The range of the first state space was set from –5 to 5 mmHg and the second from –2 to 2%. Using NGnet, the state value function $V(s(t))$ in the critic and actor value function $a_m(s(t))$ for the $m^{th}$ muscle in the actor are represented as follows:

$$V(s(t)) = \sum_{k=1}^{K} w_k^V b_k(s(t)), \tag{31}$$

$$a_m(s(t)) = \sum_{k=1}^{K} w_k^a b_k(s(t)). \tag{32}$$

Here, $b_k(s(t))$ denotes the base function represented by the following equation:

$$b_k(s(t)) = \frac{B_k(s(t))}{\sum_{l=1}^{K} B_l(s(t))}, B_k(s(t)) = exp\left[-\sum_{i=1}^{n}\left(\frac{s_i(t) - c_i}{\sigma_b^i}\right)^2\right]. \tag{33}$$

$c_i$ denotes the coordinates ($s_1, s_2$) of the center of the activation function, The parameter $\sigma_b^i, K$, and $n$ represent a constant, the number of base functions, and the number of states $s(t)$, respectively. The number of base functions $K$ was set to 144.

In an environment where the $CO_2$ percentage in the mouth ranges from 0 to 2% and the partial pressure of $CO_2$ in the blood ranges from 35 to 45 mmHg, the agent observes the

current state $s(t)$. This state includes the partial pressure of $CO_2$ in the blood $p_c$ obtained from the respiratory and circulatory systems, and the $CO_2$ percentage in the mouth (ranging randomly from 0 to 2% with 0.1 intervals) at the start of the simulation. The agent then determines the activation level input $u_m$ for the diaphragm and abdominal muscles to control lung volume using Eq (38) voluntarily. The agent receives the reward $r(t)$ as described by Eq (34).

$$r = 2.0 \left( \exp \left( -\left( \frac{\text{error}_{\text{pco2max}}}{\sigma_r} \right)^2 \right) + \exp \left( -\left( \frac{\text{error}_{\text{pco2min}}}{\sigma_r} \right)^2 \right) - 0.5 \right) - c \sum_{m=1}^{N} u_m^2, \quad (34)$$

where $error_{pco2max}$ is the difference between the maximum calculated $p_c$ and 45 mmHg, and $error_{pco2min}$ is the difference between the minimum calculated $p_c$ and 35 mmHg. Here, $\sigma_r = 6$, and $c = 0.1$. This reward function is used because the partial pressure of $CO_2$ in the blood exhibits minimal variation compared to changes in BR and HR, and blood oxygen partial pressure in normal human subjects [75].

The critic network computes the value function $V(s(t))$ from the current state $s(t)$ using Eq (31) and aims to minimize the prediction error, specifically, the TD error $\delta(t)$ as defined by Eq (35).

$$\begin{aligned} \delta(t) &= r(s(t)) + \gamma V(s(t+1)) - V(s(t)) \\ &= r(s(t)) + \left( 1 - \frac{\Delta t}{\tau} \right) V(s(t+1)) - V(s(t)). \end{aligned} \quad (35)$$

$\gamma$ denotes the discount factor ($0 \leq \gamma \leq 1$) and $\tau$ denotes the time constant of the evaluation. In the calculation of the TD error $\delta(t)$ using Eq (35) with online learning, which updates at each time step, the backward Euler approximation of the time derivative $\dot{V}(s(t))$ is often employed. This involves the eligibility trace $e_k(t)$ updated using Eq (36) [74].

$$\dot{e}_k(t) = -\frac{1}{\kappa} e_k(t) + \frac{\partial V(s(t))}{\partial w_k^V}, \quad (36)$$

where $\kappa$ denotes the eligibility trace time constant. The value function $V(s(t))$ is updated using Eq (37), incorporating the eligibility trace $e_k(t)$.

$$\Delta V(s(t)) = \alpha_V \delta(t) e_k(t) \quad (37)$$

where $\alpha_V$ denotes the learning rate of the critic. The TD error $\delta(t)$ is then computed using Eq (35).

The actor–network computes the action value function $a_m(s(t))$ for the $m^{th}$ muscle from the current state $s(t)$ using Eq (32). It learns to enhance the value function $V(s(t))$, and maximizes the expected cumulative reward. In calculating $a_m(s(t))$ via Eq (32), the weight $w_k^a$ is updated using Eq (39), incorporating the TD error $\delta(t)$. The activation level $u_m(t)$, for the $m^{th}$ muscle, is derived from Eq (38), which includes the weight of the action value function $a_m(s(t))$,

specifically, $w_k^a$.

$$u_m(t) = u_m^{max} sig(-A(\sum_{k=1}^{K}(w_k^a)_m b_k(s(t)) + \exp(-0.5V(s(t)))n_m(t)) - B), \quad (38)$$

$$\Delta(w_k^a)_m = \alpha_a \delta(t)n_m(t)\exp(V(s(t))b_k(s(t)), \quad (39)$$

$$u_m^{max} = \begin{cases} u_{ab} = 0.0, u_{di} = \text{UMAX} * sig(-5.0V_L + 5.0), & (\textit{for inspiration}) \\ u_{ab} = \text{UMAX} * sig(-5.0V_L + 5.0), u_{di} = 0.0, & (\textit{for expiration}) \end{cases} \quad (40)$$

where $u_m^{max}$ represents the maximum activation level of the $m^{th}$ muscle; $m$ = 1 or 2. $m_1$ and $m_2$ correspond to the diaphragm and abdominal muscles, respectively; $u_m^{max}$ is obtained using Eq (40), and ranges from 0 to 1. The sigmoid function is denoted by $sig()$, with constants $A$ and $B$. The actor's learning rate is $\alpha_a$. The white noise function $n_m(t)$ is randomly determined for each muscle $m$ from zero to one at each time step to explore the control output. Parameters A, B, $\tau$, $\kappa$, $\alpha_V$, and $\alpha_a$ are set to 1.0, 0.0, 0.053, 0.053, 0.3, and 0.1, respectively. UMAX determines the upper limit of voluntary activation, as described in the Results section.

Four control parameters—UMAXE, TML, EXGAIN, and PCO2—were incorporated to represent the BR, TV, and MV derived from experimental exercise data. UMAXE simulates hyperventilation at exercise onset, aligning with the central command hypothesis [76]. TML, used in Eq (24), muscle length change afferent feedback based on the peripheral neural reflex hypothesis [76], directly inputting to the RVLM to increase BR and HR via RVLM presympathetic neuron activation [77]. EXGAIN and PCO2, as defined in Eq (9), correspond to excitatory signals from NTS Chemo to RTN and pre–I/I, introduced as humoral inputs essential for initiating hyperventilation at exercise onset.

## Experimental design

Simulations were conducted for 50 s to predict the responses of respiratory and circulatory system using the specified model. Before these simulations, the four breathing control parameters—PONS, RAMPI, APSR, and UMAX—were determined. Involuntary breathing simulations were then performed for 100 s to stabilize the breathing conditions. Subsequently, simulations utilizing the online ACRL were executed with a time step of 0.0001 s under three conditions:

1. Active or passive STEs with both knees flexed as per Ishida et al. experimental setup [1];

2. VB control to adjust the breathing rate from 6 to 14 bpm;

3. and Physiological changes under MS loads.

The simulation conditions commenced 20 s after the start due to the initial respiratory state instability observed until approximately 15 s. The model, incorporating the closed–loop system by Molkov et al. [64], autonomously regulates respiration akin to living organisms. Consequently, after inputting the initial four parameters, approximately three respiratory cycles are required to achieve stable respiration at rest. The parameters for controlling respiratory states under the three conditions are detailed in Table 1. In Condition 1, the BR was 14.5 bpm, which represents the BR before STEs, almost consistent with experimental data [1]. PONS, RAMPI, and APSR adjusting automatic/involuntary breathing; UMAX adjusts VB; UMAXE, TML, EXGAIN, and PCO2 adjust respiratory states during STE.

The computational model of the respiratory–circulatory system as shown in Figs 8 and 9 was implemented using Python 3.8. All simulations were executed on a Dell OptiPlex 5050 computer equipped with an Intel Core i7–6700 and 16 GB Memory (DDR4).

## Acknowledgments

This research was conducted as part of a collaborative project between the Intelligent Mobility Society Design/Social Cooperation Program and the Next Generation Artificial Intelligence Research Center at the University of Tokyo, and Toyota Central R&D Labs Inc. We acknowledge Editage (www.editage.com) for their assistance with English language editing.

## Author Contributions

**Conceptualization:** Masami Iwamoto.

**Data curation:** Masami Iwamoto, Satoko Hirabayashi.

**Formal analysis:** Masami Iwamoto, Satoko Hirabayashi, Noritoshi Atsumi.

**Funding acquisition:** Masami Iwamoto.

**Investigation:** Masami Iwamoto, Satoko Hirabayashi, Noritoshi Atsumi.

**Methodology:** Masami Iwamoto, Satoko Hirabayashi, Noritoshi Atsumi.

**Project administration:** Masami Iwamoto.

**Resources:** Masami Iwamoto.

**Software:** Masami Iwamoto, Satoko Hirabayashi, Noritoshi Atsumi.

**Supervision:** Masami Iwamoto.

**Validation:** Masami Iwamoto, Satoko Hirabayashi, Noritoshi Atsumi.

**Visualization:** Masami Iwamoto, Satoko Hirabayashi, Noritoshi Atsumi.

**Writing – original draft:** Masami Iwamoto.

**Writing – review & editing:** Masami Iwamoto, Satoko Hirabayashi, Noritoshi Atsumi.

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
