## [Decision Letter · Decision Letter 0]

16 Nov 2024

Dear Dr. Iwamoto,

We are pleased to inform you that your manuscript 'In−silico simultaneous respiratory and circulatory measurement during voluntary breathing, exercise, and mental stress: A computational approach' has been provisionally accepted for publication in PLOS Computational Biology.

Best regards,

Lyle Graham

Section Editor

PLOS Computational Biology

Feilim Mac Gabhann

Editor-in-Chief

PLOS Computational Biology

Jason Papin

Editor-in-Chief

PLOS Computational Biology

Reviewer's Responses to Questions

**Comments to the Authors:**

Reviewer #1: Thank you for addressing my concerns in your revision of the manuscript. I think you have greatly improved the work and I believe it to be a significant contribution to the computational modeling of autonomic function.

**Have the authors made all data and (if applicable) computational code underlying the findings in their manuscript fully available?**

Reviewer #1: Yes

PLOS authors have the option to publish the peer review history of their article (what does this mean?). If published, this will include your full peer review and any attached files.

Reviewer #1: **Yes: **Christopher G. Wilson

---

## [Editor Report · Acceptance letter]

26 Nov 2024

PCOMPBIOL-D-24-01602 

In−silico simultaneous respiratory and circulatory measurement during voluntary breathing, exercise, and mental stress: A computational approach

Dear Dr Iwamoto,

I am pleased to inform you that your manuscript has been formally accepted for publication in PLOS Computational Biology. Your manuscript is now with our production department and you will be notified of the publication date in due course.

With kind regards,

Zsofia Freund
